# Different influences of moral violation with and without physical impurity on face processing: An event-related potentials study

Siyu Jiang ◉, Ming Peng ◉*, Xiaohui Wang

Key Laboratory of Adolescent Cyberpsychology and Behavior of the Ministry of Education and School of Psychology, Central China Normal University, Wuhan, China

◉ These authors contributed equally to this work.
* pengm2015@mail.ccnu.edu.cn

**Data Availability Statement:** All relevant data are within the paper and its Supporting information files.

## Abstract

It has been widely accepted that moral violations that involve impurity (such as spitting in public) induce the emotion of disgust, but there has been a debate about whether moral violations that do not involve impurity (such as swearing in public) also induce the same emotion. The answer to this question may have implication for understanding where morality comes from and how people make moral judgments. This study aimed to compared the neural mechanisms underlying two kinds of moral violation by using an affective priming task to test the effect of sentences depicting moral violation behaviors with and without physical impurity on subsequent detection of disgusted faces in a visual search task. After reading each sentence, participants completed the face search task. Behavioral and electrophysiological (event-related potential, or ERP) indices of affective priming (P2, N400, LPP) and attention allocation (N2pc) were analyzed. Results of behavioral data and ERP data showed that moral violations both with and without impurity promoted the detection of disgusted faces (RT, N2pc); moral violations without impurity impeded the detection of neutral faces (N400). No priming effect was found on P2 and LPP. The results suggest both types of moral violation influenced the processing of disgusted faces and neutral faces, but the neural activity with temporal characteristics was different.

## Introduction

### Moral judgment and disgust emotion

Moral violation behavior is one example of a behavior against social norms [1]. How do people judge whether a behavior is a moral violation in daily life? Psychologists have been paying attention to this problem for a long time [2,3]. The study of people's moral judgment can help us understand how social norms are formulated and how social order is maintained [4–7]. Several theories have tried to explain moral judgment using the concepts of moral reasoning (e.g., Piaget's moral cognitive development model) [2,3], moral emotion (e.g., the social intuitionist model) [8,9], or both (e.g., dual-processing theory) [10]. Both the social intuitionist model and

**Funding:** This research was supported by the MOE (Ministry of Education in China) Project of Humanities and Social Sciences (No.18YJC90018), self-determined research funds of CCNU from the colleges' basic research and operation of MOE (CCNU19TD018), and Fundamental Research Funds for the Central Universities (CCNU20QN024). The funders had no role in study design, data collection and analysis, decision to publish, or preparation of the manuscript.

**Competing interests:** The authors have declared that no competing interests exist.

dual-processing theory emphasize that moral emotions, such as disgust, anger and empathy, play an important role in moral judgment. Specifically, people make more extreme judgments about others' moral behavior when they feel strong emotion. However, there has been controversy about judgments of moral behavior and the specific emotion of disgust.

The primary function of disgust is to prevent contact with pathogens and encourage avoidance of things that may carry pathogens [11–14]. Disgust are often induced by excrement, rotten food, blood, certain sexual behaviors, bodily wounds, illness, an unhealthy body [11,15]. Disgust also makes people pay selective attention to stimuli that heuristically connote the presence of diseases, such as anomalous face, the obese [16], and the elderly [17–19]. Moreover, people may also respond with disgust when others violate moral standards. Moral violations involving impurities can cause the spread of pathogens; this belief induces the emotion of disgust [9,20–22]. However, not all moral violations involve impurities. There is still controversy over whether moral violations without impurities also induce the emotion of disgust. The answer to this question may have implication for understanding where morality comes from and how people make moral judgments [23–25].

## The emotion of disgust and moral violations with and without impurity

Some researchers have considered disgust to be a specific reaction to violations of purity or sanctity [20,21]. They proposed the CAD triad hypothesis, which links anger to autonomy (individual rights violations), contempt to community (violation of communal codes, including hierarchy), and disgust to divinity (violations of purity-sanctity) [21]. They asserted that people say they are "disgusted" at harm (e.g. demeaning offences) or unfairness only because of semantic confusion with other emotions, such as anger [26]. Rozin et al. [21] asked participants to read sentences depicting different kinds of moral violations, and to describe each sentence with an emotional word or an emotional face. Participants used the word "disgust" or a disgusted face more often when describing moral violations with physical impurity (such as "a person is watching someone as he/she bites into an apple with a worm in it") than moral violations without physical impurity (such as "a person is seeing someone steal a purse from a blind person."). Horberg, et al. [20] found similar results by using vignettes rather than sentences as stimuli.

However, other researchers have speculated that disgust may arise when a person is seen as having a fundamentally bad character, regardless of what type of moral norm they have violated [15,27–30]. They found that disgust can also be triggered by behaviors that violate human dignity [31], and in some cases justice [32]—can help prevent further victimization [23]. For example, participants used "disgust" and "anger" to express their feelings in response to various social moral violations [33–35]. When the participants accepted an unfair distribution, a moral violation without physical impurity, they showed a disgusted facial expression, i.e., obvious facial levator activity [33,34]. In another study, interpersonally unfair treatment at work, defined as treatment that violated an individual's sense of dignity and respect, triggered the emotion of disgust [35]. The results of these studies suggest that disgust can be induced not only by moral violations with physical impurity, but also those without physical impurity.

**Neural mechanisms underlying moral violations with and without impurity.** It's rather difficult to make it clear that whether disgusted emotion induced by moral violations with impurity was the same as moral violations without impurity by using verbal report [33]. However, it could be better to combine neuropsychological techniques, e.g. fMRI or ERPs, to reveal their neural mechanisms to get what happened underlying different kinds of moral violation. Several studies have been conducted using functional magnetic resonance imaging (fMRI) to examine the biological mechanisms of two kinds of moral violations. Sanfey et al. [32] showed that when the participants encountered unfair distribution in an ultimatum game, the anterior

insula, dorsolateral prefrontal lobe and anterior cingulate gyrus were significantly activated. The recruitment of similar neural structures, namely the anterior insula, in both purity and moral violation. Moll et al. [36] compared brain regions that were activated when participants were processing pure disgust (physical violation without moral violation) and moral disgust (moral violation without physical impurity). They found that two kinds of violation recruited overlapping neural substrates, such as medial orbitofrontal; and distinct brain regions, mainly in the frontal and temporal lobes.

Studies using ERP data have also been helpful in identifying the time course of two kinds of moral violation. Yang and colleagues [23,24,37] conducted a series of ERP studies to examine this issue. In one study [37], they used a Go/No-Go paradigm to evoke lateralized readiness potentials (LRPs) and found that moral information was processed prior to physical disgust. In another study [23], participants were asked to judge the acceptability of different types of behaviors that varied in their level of moral wrongness (moral vs. immoral) and physical disgust (with or without physical disgust). They found that immoral behaviors with or without physical disgust elicited greater amplitudes of 300–400 ms and 500–600 ms at frontal sites than morally neutral yet physically disgusting behaviors. These researchers also used the recognition potential (RP) of 200-300ms to investigate how individual differences in moral preferences regulate moral or disgust words processing in the pre-semantic stage [24]. The results showed that participants who had lower scores in the harm and care related moral violations showed significantly larger RP amplitudes for disgust words than for neutral words. These studies demonstrated moral violation behavior and physically disgusting behavior were differed in the time course of neural activity. However, there was no consistent conclusion on whether there is a difference between moral violation behavior with and without physically disgusting characteristics. In this study, we explored the neural mechanism underlying these two kinds of moral violation by compared their different influences on subsequent social information processing.

## Emotion induced by moral violations influence subsequent processing

According to social intuitionist model [8], moral violations can trigger moral emotions, which can directly influence people's behavior and judgment [38–40]. There is some evidence to support this model with regard to the moral emotion of disgust. For example, in one study participants who read about a moral violation found a drink to be more disgusting than other participants who were given the same drink; participants in the moral violation condition were also less likely to drink the disgusting beverage [41]. However, no studies to date have examined whether different kinds of moral violations have the similar effect on subsequent emotion processing. This research gap provides an opportunity to further explore the different neural mechanisms underlying moral violations. The current experiment was an affective priming study in which emotion, especially disgust, was assessed in terms of its influence on the processing of social information.

In affective priming studies [42–45], participants are presented with background information (pictures, words, sentences), and then presented with faces showing different affective expressions. Facilitated processing (reaction times) is showed in affectively congruent targets and impaired processing is showed in incongruent ones, a phenomenon known as the affective priming effect. The presentation of priming stimuli can automatically activate the related emotional representations of the brain and carry out implicit emotional processing [46,47]. Multiple studies have shown the priming effect, and this effect can be also detected using ERP data [45,48,49].

Three ERP components, P2, N400, and LPP are of interest in affective priming tasks [48–51]. P2, which usually peaks around 200–250 ms, is located over the centro-frontal and the parieto-occipital region. It represents some aspects of higher-order perceptual processing, modulated by attention to visual stimuli [52]. N400 is a negative deflection observed around 400 ms after target presentation at centro-parietal scalp electrodes. It is sensitive to semantic relatedness and congruency [48]. LPP is a central-parietal, midline component that becomes evident after 300 ms following the presentation of emotional stimuli onset and can be increased for several seconds [53]. It has been associated with emotional processing of faces and appraisal of affective meaning [54,55]. These components have shown sensitivity to the effects of affective priming on the visual processing of emotional faces [49–51,56,57]. Compared with the target faces with inconsistent background information emotion, the target faces with consistent background information emotion elicited larger P2 amplitude in frontal lobe [49,50]. On the contrary, when compared with emotionally consistent priming background-target pairs, inconsistent priming background-target pairs elicited larger N400 and LPP amplitudes [51,56,57].

## Overview of this study

In this study, we employed a priming paradigm in which we presented sentences about moral violations and sentences about neutral behaviors as the priming materials before participants were asked to detect faces that expressed disgust. Participants' reaction times and ERP data were collected to test whether moral violation behaviors with and without physical impurity had the same influence on subsequent face processing.

Compared with a recognition task that presents a single face [48,56], the face search task is more sensitive in detecting early perceptual processing [51,52]. For example, two studies revealed that when a negative face was the target and a neutral face served as the distractor, an N2pc wave was elicited in the early stage of information processing [58,59]. N2pc was also observed over occipital scalp electrodes in the time range of 180–300 ms after stimulus onset contralateral to the side of an attended visual event [60]. It is generally believed that N2pc is closely related to spatial selective attention [61], and N2pc is often used as an indicator of attention allocation to target stimuli [62]. Therefore, this study used the face search task to investigate the P2, N400, LPP and N2pc amplitudes elicited by faces after participants were primed with sentences about moral violations with impurity, moral violations without impurity, and neutral behaviors.

Based on previous behavioral [33–35] and ERP studies [23], we hypothesized that moral violations both with and without physical impurity will influence the processing of disgusted faces; however, the two types of moral violation will elicit neural activity with different temporal characteristics. First, sentences depicting moral violations with impurity include words that can elicit immediate disgust, and this effect should be registered in the early stage of face processing. This can be indicated by larger P2 and smaller N2pc amplitudes induced by a disgusted face after sentences depicting moral violations with impurity compared to sentences depicting neutral behaviors. Second, moral violations without impurity involve greater semantic complexity and require greater resources for semantic integration [63,64], and these violations should influence face processing in the late stage, indicated by N400 and LPP amplitudes. Larger N400 and LPP induced by neutral faces, or smaller N400 and LPP induced by disgusted faces, will be observed after sentences depicting moral violations without impurity compared to sentences depicting neutral behaviors.

## Method

### Participants

We recruited thirty right-handed participants from a university in central China in the study. Data from two participants were excluded from all data analyses because the accuracy of the behavioral data was less than 80%. Data from three participants were excluded from the ERP data analyzes due to intensive head movements during EEG recording ($N = 2$) or excessive artifacts ($N = 1$). The EEG data from 25 participants were analyzed (12 males; mean age = 21.16 ± 0.33 years). Using this sample size ($N = 25$) and a pre-defined effect size ($\eta_p^2$) of 0.25, a power analysis in G*Power [65] showed that this sample size would give over 90% power to detect an effect.

All participants were native Chinese speakers and had normal or corrected-to-normal vision. None of the participants had a history of neurological or psychiatric disorders. All participants signed an informed consent form for the experiment and were paid 40 RMB for their participation. The informed consent form and the study were reviewed and approved by the ethics committee of the Faculty of Psychology, Central China Normal University, China.

### Materials

Sentence materials: The sentence materials were developed in prior research [23,24,66]. Because the word order in some sentences was slightly different from expressions in daily life, we modified the wording of some sentences before the ERP experiment (e.g., "At a public swimming pool, a person is shitting" was revised into "A person is shitting at a public swimming pool"). Thereafter, 42 neutral sentences (NN) (e.g., "A person is buying some daily necessities in the market"), 42 moral violation sentences (WN) that have nothing to do with physical impurity (e.g., "A person is swearing loudly in public"), and 42 moral violation sentences (WD) that are related to physical impurity (e.g., "A person is spitting in public") were used in the ERP experiment. The key words in the sentences depicting moral violations with impurity were related to physical impurity were related to excrement, blood, wounds and sickness. Because sexual stimulation could activate additional brain areas [67], the sentences depicting moral violations with impurity did not include sexual behaviors.

The 42 sentences of each type (NN, WN, WD) were repeated five times, and addition 14 randomly selected sentences of each type were repeated one more time. In all, 672 sentences were presented to each participant (42*5*3 + 14*3).

A separate set of participants who did not take part in the ERP experiment were asked to rate the degree to which each sentence was morally wrong or disgusting. The 64 participants were randomly assigned to two groups (moral group, disgust group); each group had 32 subjects. Ratings were made on a 9-point scale, either (1 = very immoral, 9 = very moral) or (1 = not disgusting at all, 9 = very disgusting). The results showed that the main effect of sentence types was significant at moral score ($F (2,62) = 677.986$, $p < 0.001$, $n_p^2 = 0.956$, NN = 337.00 ± 9.12, WN = 72.094 ± 4.47, WD = 64.41 ± 3.44). Post hoc multiple comparisons revealed that WN < NN, $p < 0.001$; WD < NN, $p < 0.001$; there was no significant difference between WD and WN, $p = 0.113$. The main effect of sentence types was significant at disgust score ($F (2,62) = 839.187$, $p < 0.001$, $n_p^2 = 0.964$, NN = 47.13 ± 1.93, WN = 308.50 ± 7.99, WD = 331.72 ± 8.63). Post hoc multiple comparisons revealed that WN > NN, $p < 0.001$; WD > NN, $p < 0.001$; WD > WN, $p < 0.001$.

Face materials: Images of faces expressing neutral emotion and disgust were selected from the Chinese Affective Facial Picture System (CAFPS) [68]. All pictures were black and white images, with a resolution of 260 × 300. There were 30 faces expressing neutral emotion and 30

faces expressing disgust. Half the faces were male, and half female. The 9-point score of emotional intensity of images provided by the CAFPS showed that the emotional intensity of disgust faces (6.42 ± 0.65) was significantly higher than that of neutral faces (5.78 ± 0.17), $F$ (1, 58) = 27.54, $p < 0.001$, $\eta_p^2 = 0.322$.

## Procedure

The participants sat in a quiet room and adjusted their body posture appropriately. They were told to control head movement as much as possible after the start of the experiment, and not to blink frequently. During the experiment, the participants were asked to complete the reaction time task. The stimulus display and behavioral data acquisition were programmed using E-Prime software.

Participants were first presented with a practice block of 6 trials to familiarize them with the task, and then the formal experiment and EEG recording began. Every trial started with a central fixation for 500 ms, followed by a blank screen with a random duration between 400 ms and 600 ms. Then the sentence stimulus was presented at the center of the screen for 2000 ms (the participants were asked to read the sentence silently), followed by a blank screen with a random duration between 500 ms and 800 ms. The two facial expression stimuli were then presented at the center of the screen. On half of the trials, one face described disgust, the other was neutral, and on the other half, both faces were neutral. The participants' task was to judge whether one of the two faces in the picture expressed disgust. The trial ended with a blank screen with duration of 500 ms (Fig 1). The F and J buttons as indicators of "yes" and "no" were counterbalanced across participants. The presentation of the sentence ended when the participant pressed one of these keys. The experiment was divided into six blocks and 112 trials for each block. After each block, a rest screen appeared that allowed subjects to have a short break.

## ERP recording and data analysis

EEG was recorded from 64 scalp positions using tin electrodes mounted in an elastic cap. All inter-electrode impedance was maintained below 5 kΩ. The reference electrode set at ref in the positive center. The ground point set at the midpoint of Fpz and Fz (GND). The vertical electrooculograms (VEOG) were recorded using one electrode placed below the right eye. All signals were sampled at 500 Hz and band-pass filtered within a 0.05~100 Hz frequency range. The behavioral data were recorded synchronously when EEG signals were collected.

EEG data were analyzed using the Matlab R2013b. Averaging of ERPs for the six conditions was computed off-line. All EEG signals were re-referenced to the average of the left and right

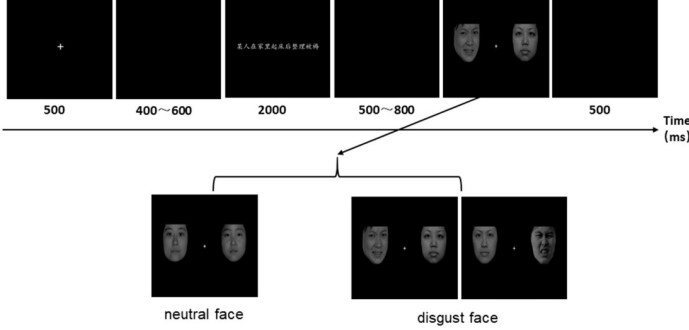

**Fig 1. The schematic illustration of the experimental procedure.**

mastoids (Tp9 and Tp10). The data were filtered offline using a 0.01~35 Hz band-pass infinite impulse response (IIR) filter. The independent component analysis was used to reject the eye movement artifacts (blinks and eye movements). Trials contaminated by artifacts exceeding a threshold of±100 μV were eliminated. ERP waveforms were time-locked to the onset of the facial expressions.

The averaged epoch for ERPs was 1200 ms, including a 200 ms pre-stimulus baseline. According to the observation and analysis of the previous study and the general average map [23,25,40,63], P2 (170–220 ms), N400 (360–460 ms), LPP (700–100 ms) and N2pc (160–300 ms) were selected as time-window of analysis. C1, CZ, C2, CP1, CPZ and CP2 electrodes were selected to analyze the average amplitude of P2, N400 and LPP. Analyses on N2pc average amplitude focused on lateral posterior electrodes PO7 and PO8, where the N2pc component is maximal.

The waveforms of disgust faces at the contralateral and ipsilateral electrodes were averaged. The ipsilateral waveform (average of voltage at the left-sided electrode for the left visual field target and voltage at the right-sided electrode for the right visual field target) and contralateral waveform (average of voltage at the left-sided electrode for the right visual field target and voltage at the right-sided electrode for the left visual field target) time-locked to the visual display for three sentence conditions at PO7/PO8 electrode sites were computed separately.

The degrees of freedom for the F-ratio were corrected according to the Greenhouse-Geisser method. For post hoc analysis, the Bonferroni correction was used.

## Results

### Behavioral results

We conducted 3 (sentence types: NN, WN, WD) by 2 (facial types: disgust face, neutral face) Repeated-measures ANOVAs on reaction times (RTs). The main effect of sentence types was not significant ($p = 0.710$). The main effect of facial expression types was significant ($F (1,27) = 13.176$, $p = 0.001$, $\eta_p^2 = 0.328$). RT for disgust facial expressions (798.465 ± 30.198 ms) was faster than for neutral facial expressions (861.982 ± 36.004 ms), $p = 0.001$. The interaction between sentence types and facial expression types was significant ($F (2,54) = 24.004$, $p < 0.001$, $\eta_p^2 = 0.471$). A significant sentence type effect was observed in the disgust facial expression conditions ($F (2,54) = 18.65$, $p < 0.001$; NN: 818.134 ± 30.922 ms, WN: 799.431 ± 30.659 ms, WD: 777.829 ± 29.720 ms; WD < NN, $p < 0.001$; WD < WN, $p = 0.001$; WN < NN, $p = 0.039$). A significant sentence type effect was observed in the neutral facial expression conditions ($F (2,54) = 10.28$, $p < 0.001$; NN: 857.469 ± 34.953 ms, WN: 868.957 ± 35.306ms, WD: 890.434 ± 37.915ms; NN < WD, $p < 0.001$; WN < WD, $p = 0.029$; NN = WN, $p = 0.32$) (Fig 2).

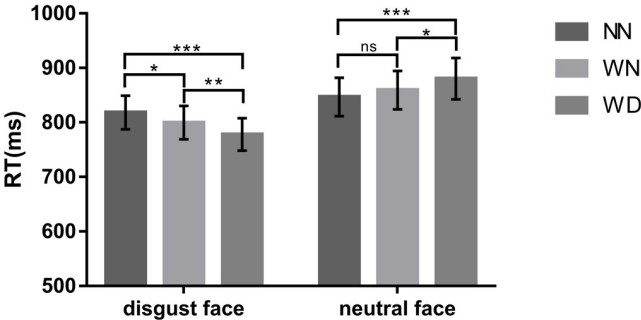

**Fig 2. The reaction time of face search task across the NN, WN, and WD priming conditions.** NN: neutral sentences. WN: moral violation sentences that have nothing to do with physical impurity. WD: moral violation sentences that are related to physical impurity. "*" means $p < 0.05$, "**" means $p < 0.01$, "***" means $p < 0.001$, "ns" means $p > 0.05$. Error bars: +/− 1 SE.

## ERP results

**P2 (170~220ms).** We conducted 3 (sentence types: NN, WN, WD) by 2 (facial types: disgust face, neutral face) repeated measures ANOVA for the P2. The ANOVA results showed that the main effect of sentence types was not significant ($F(2,48) = 1.388$, $p = 0.259$, $\eta_p^2 = 0.055$), the main effect of facial types was not significant ($F(1,24) = 1.035$, $p = 0.319$, $\eta_p^2 = 0.041$), the interaction between sentence types and facial types was not significant ($F(2,48) = 0.662$, $p = 0.520$, $\eta_p^2 = 0.027$).

**N2pc (160~300ms).** We conducted repeated measures ANOVAs on N2pc different waves (the contralateral waveform minus the ipsilateral waveform) with sentence types as a within-subject factor. The ANOVA results showed that the main effect of sentence types was significant ($F(2,48) = 7.032$, $p = 0.002$, $\eta_p^2 = 0.227$), under the NN priming condition, disgust face elicited larger amplitudes, compared with the WD priming condition (-1.114 vs. -0.708, $p = 0.005$) and the WN priming condition (-1.064 vs. -0.806, $p = 0.072$). No significant difference was observed between the WN priming condition and the WD priming condition (-0.806 vs. -0.708, $p = 0.904$) (Figs 3 and 4).

**N400(360~460ms).** We conducted 3 (sentence types: NN, WN, WD) by 2 (facial types: disgust face, neutral face) repeated measures ANOVA for the N400. The ANOVA results showed that the main effect of sentence types was not significant ($p = 0.228$). The ANOVA results showed that the main effect of facial types was significant ($F(1,24) = 6.460$, $p = 0.018$, $\eta_p^2 = 0.212$), the neutral faces (4.121 ± 0.948μV) elicited larger amplitudes than the disgust faces (4.908 ± 0.989μV). The interaction between sentence types and facial types was significant ($F(2,48) = 3.042$, $p = 0.012$, $\eta_p^2 = 0.167$). Additional simple effects analyses revealed that sentence type effect was significant in the neutral facial conditions ($F(2,48) = 6.55$, $p = 0.003$). Neutral faces elicited larger amplitudes under the WN priming condition (3.686 ± 0.979μV), compared with the NN priming condition (4.520 ± 0.927μV), $p = 0.009$. There was no significant difference between the WN priming condition (3.686 ± 0.979μV) and the WD priming condition (4.156 ± 0.963μV) ($p = 0.18$). There was no significant difference between the WN priming condition and the NN priming condition ($p = 0.24$). There was no sentence type effect in the disgust faces conditions ($p = 1.00$) (Figs 5–7).

**LPP (700~1000ms).** We conducted 3 (sentence types: NN, WN, WD) by 2 (facial types: disgust face, neutral face) repeated measures ANOVA for the LPP. The ANOVA results showed that the main effect of sentence types was not significant ($F(2,48) = 0.404$, $p = 0.670$, $\eta_p^2 = 0.017$), the main effect of facial types was not significant ($F(1,24) = 0.048$, $p = 0.828$, $\eta_p^2 = $

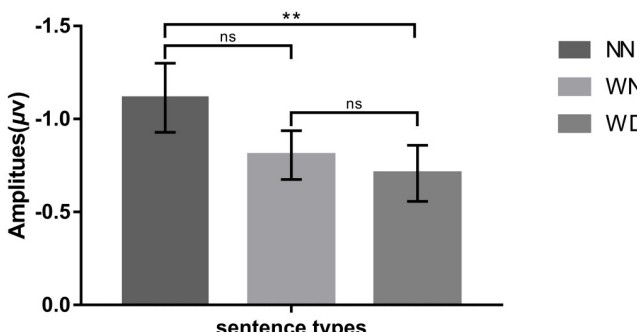

**Fig 3. Bar graphs of the N2pc difference amplitudes in response to disgust face across the NN, WN, and WD priming conditions.** "*" means $p < 0.05$, "**" means $p < 0.01$, "***" means $p < 0.001$, "ns" means $p > 0.05$. Error bars: +/− 1 SE.

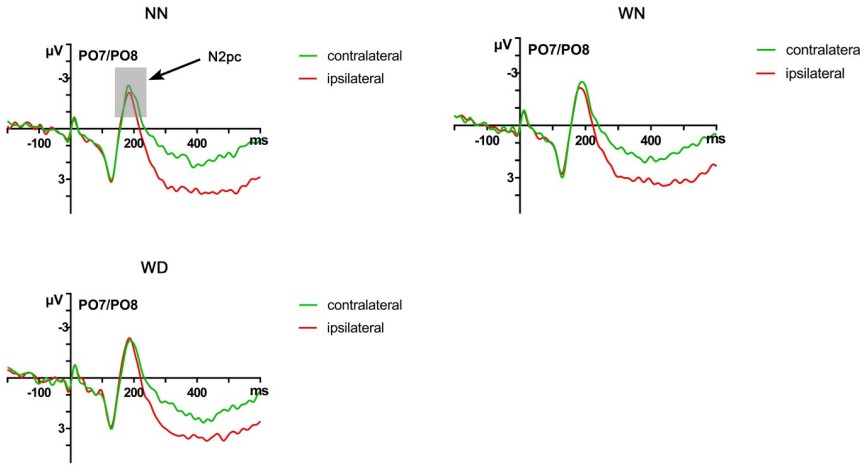

**Fig 4. Grand average ERP waveforms of target disgust face on contralateral and ipsilateral sides of PO7/PO8 across the NN, WN, and WD priming conditions.**

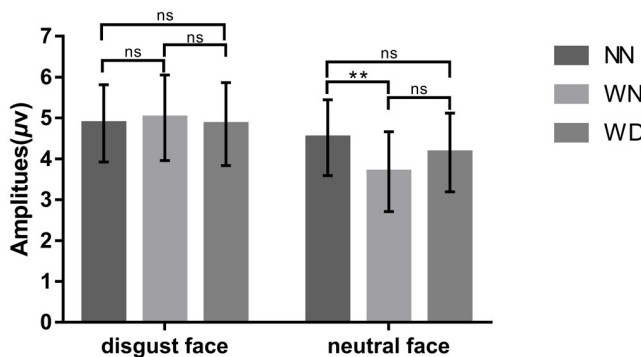

**Fig 5. Bar graphs of the N400 amplitudes in response to disgust face and neutral face across the NN, WN, and WD priming conditions.** "*" means $p < 0.05$, "**" means $p < 0.01$, "***" means $p < 0.001$, "ns" means $p > 0.05$. Error bars: +/− 1 SE.

0.002), the interaction between sentence types and facial types was significant ($F$ (2,48) = 3.736, $p = 0.031$, $\eta_p^2 = 0.135$). Additional simple effects analyses revealed that sentence type effect was significant in the neutral face condition ($F$ (2,48) = 3.80, $p = 0.029$). Post event multiple comparative analysis found that there was no significant difference between the sentence types under the disgust face condition (NN = WN, $p = 0.089$; NN = WD, $p = 1.000$; WN = WD, $p = 0.158$). There was no sentence type effect in the disgust face condition ($p = 0.469$).

## Discussion

Although moral violations with impurity have been shown to elicit disgust, it has been unclear whether moral violations without impurity elicit the same emotion. This is the first study that employed an affective priming paradigm to investigate how moral violation behaviors with and without physical impurity influence subsequent processing of faces expressing disgust or neutral emotion. This study would help us to reveal the neural mechanisms underlying both kinds of moral violation. Results of behavioral data and EEG data showed that moral violations both with and without impurity promoted the detection of disgusted faces (RT, N2pc); moral

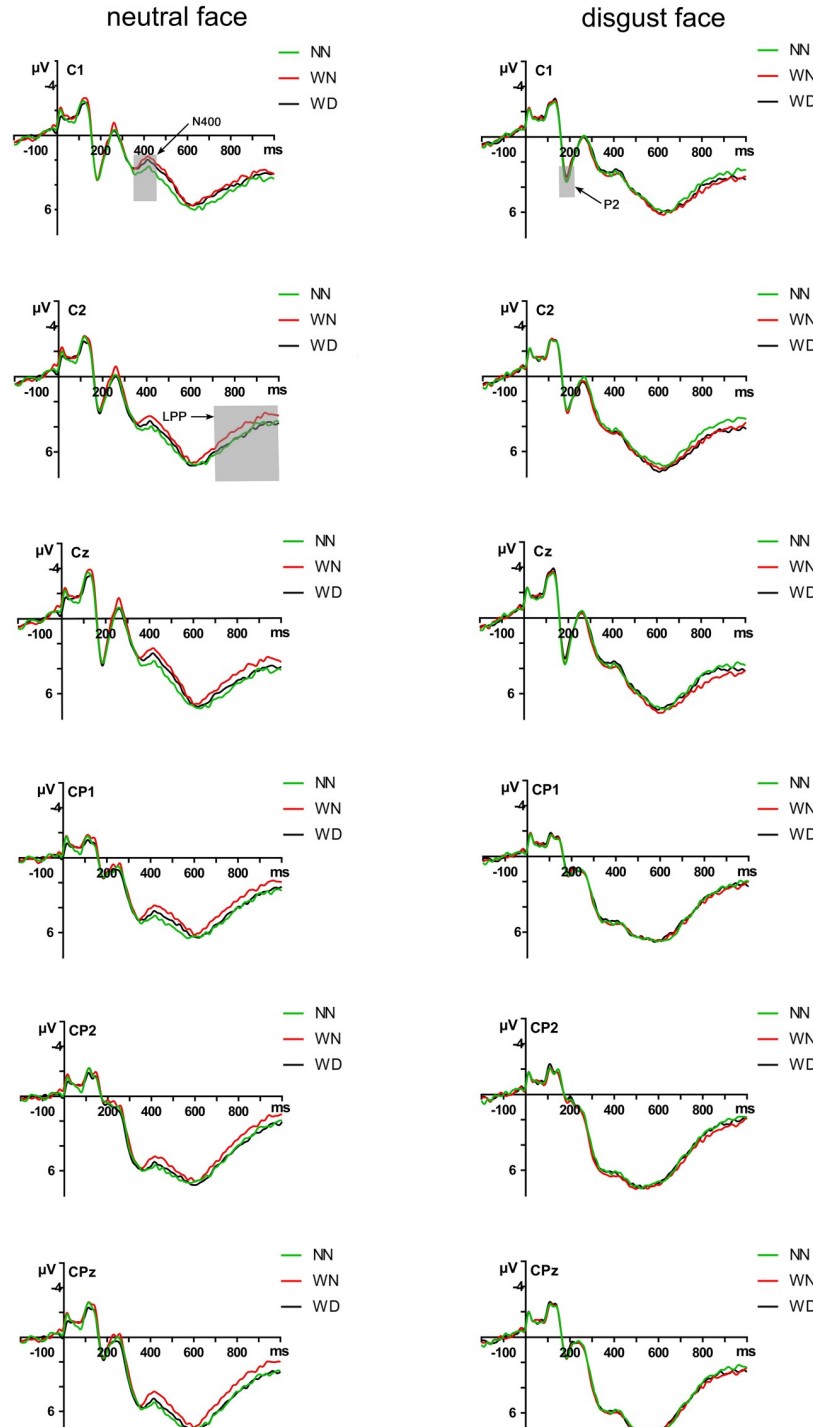

**Fig 6. Grand average ERP waveforms recorded from C1, Cz, C2, CP1, CPz and CP2 in response to disgust face and neutral face across the NN, WN, and WD priming conditions.**

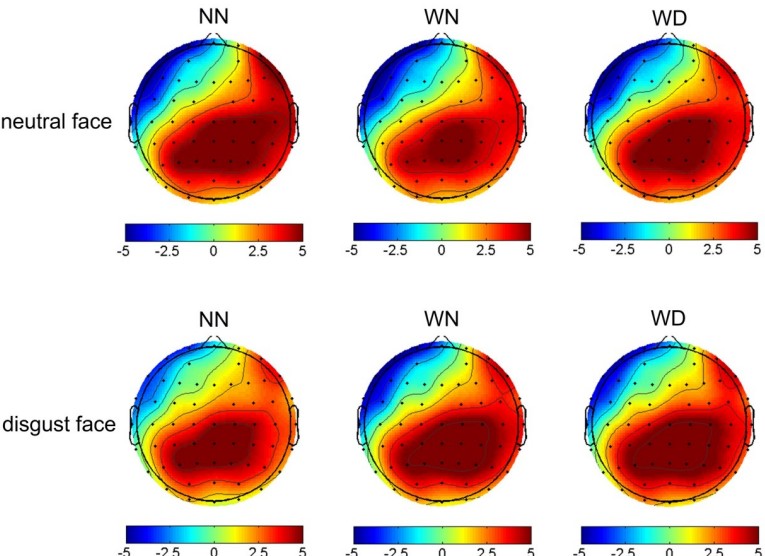

**Fig 7. Topographical maps of voltage amplitudes of N400 in response to disgust face and neutral face across the NN, WN, and WD priming conditions.**

violations without impurity impeded the detection of neutral faces in the late stage (N400). The results suggest that both types of moral violation could be elicited similar emotion, but the emotion was stronger in the case of moral violations with impurity, and more complex emotions were induced in moral violations without impurity.

In the affective priming task [42,45,69,70], facilitated processing was shown when the prime and target were affectively congruent, and impaired processing was shown when the prime and target were emotionally incongruent, that is the affective priming effect. Such an effect shows that the emotional information provided by a stimulus can be implicitly and automatically evaluated. In this study, on the one hand, participants' reaction time in detecting the disgusted faces under the priming condition of WD sentences was faster than under the priming condition of NN sentences; participants' reaction time in detecting neutral faces under the priming condition of WD sentences was slower than under the priming condition of NN sentences. These results suggest that WD, which elicited disgust, facilitated the recognition of the disgusted face and hindered the recognition of the neutral face.

On the other hand, participants' reaction time in recognizing disgusted faces under the priming condition of WN sentences was faster than under the priming condition of NN sentences; however, reaction time in recognizing neutral faces under the condition of WN sentences did not differ significantly from reaction time under the condition of NN sentences. The result showed WN sentences facilitated disgusted face processing, but not impeded neutral face processing, which suggests that WN could elicit similar emotion as WD, but the intensity was weaker than WD.

As to ERP results, consistent with the hypothesis, the N2pc results showed that both types of moral violations promote the processing of disgusted faces, but the influence was weaker for the moral violations without physical impurity. As an ERP component closely related to spatial selective attention, N2pc could be used as an indicator of attention allocation to the target stimulus [62]. Compared with WD sentences and WN sentences, NN sentences induced larger amplitudes of N2pc, suggesting that participants paid more attention to disgusted faces after reading neutral sentences. By contrast, after reading WN and WD sentences, people did not

pay more attention to disgusting faces. Thus, the detection of target faces was affected by the priming [42]. When emotion induced by the priming sentence was inconsistent with the facial emotion, more attentional resources were needed to orientate to the disgusted face, inducing larger N2pc amplitude [71]. When they were consistent, the disgusted face easily captured attention, inducing lower N2pc.

Both of moral violation with and without impurity promote the orientation to disgusting face, we inferred that disgust could be elicited both by moral violations with impurity and those without impurity. It is consistent with the results of previous studies which suggests disgust could be induced by violation of moral norm no matter whether they are related to impurity or not [33,41]. Moral violations without physical impurity may play an important role in interpersonal communication. People's disgust for this kind of behavior may be protective because they may reduce contact with individuals who could bring harm to themselves or their groups in social intercourse [31]. However, considering that this is the first study to test the influence of different moral violation behaviors on subsequent face processing, more evidence is needed to draw conclusions about this possibility.

The N400 results partly supported the hypothesis that larger N400 induced by disgusted face in WD than in NN primes. N400 amplitudes elicited by emotional incongruence were larger than those elicited by emotional congruence [63]. In the time-window of 360~460 ms, there was no significant difference in N400 amplitudes between WD and WN primes, or between WD and NN primes, which suggests the strength of emotion by WD was decrease at this stage. This result may be due to the emotional significance and complexity of sentences. Compared to the sentences describing neutral behaviors, sentences describing moral violations with impurity had more obvious information that would induce disgust. Because WD induced higher emotional arousal and could be noticed faster in the early processing stage [63], WD sentences and faces were integrated earlier. In the late stage, the emotion induced by WD sentences gradually subsided, with less impact on N400 amplitudes.

In contrast, consistent with the hypothesis, we found that neutral faces induced larger N400 amplitudes under the priming condition of WN sentences, compared with the priming condition of NN sentences. Compared to WD sentences, WN sentences have been found to be more strongly related to the retrieval of social moral information and complex social evaluations (e.g., relationship to target, benefits and costs of condemnation) [14,72]. People need to deeply analyze the nature of stimuli (causal reasoning, counterfactual thinking) [14], so the semantic processing of moral violations without physical impurity may take more time. One study found that compared with scenes that did not violate social norms, scenes that violated social norms induced larger slow waves at parieto-occipital sites in late stage processing [64]. It suggested that moral violations without impurity require greater resources for semantic integration [63,64]. Therefore, based on the results of the current study we speculate that because of the complexity of processing WN sentences, there was an emotional inconsistency effect on N400.

Moral violations without impurity did not influence the processing of the emotion of disgust but did influence the processing of neutral faces in the late stage, suggesting other complex emotions might be induced by WD. Future studies can explore this issue by examining whether moral violations with and without impurity also elicit other moral emotions, such as anger.

However, P2 and LPP results were inconsistent with our hypothesis. We found no significant difference in P2 and LPP amplitudes induced by neutral faces and disgusted faces under different types of moral violation priming conditions. These results suggest that face processing was not modulated by different types of moral violation. This finding is inconsistent with what would be expected based on some affective priming studies that found that priming effect would be observed on P2 and LPP amplitudes [49,50,56].

One of the reasons for the difference we inferred is different target task used in this study. When the priming task was used in previous studies, the target task was judgment of a single face or a single word—that is, a recognition task—instead of the search task used in our study [48,56]. Whereas face searching tasks emphasize the attentional orientation or how attention was captured by emotion [73], recognition tasks focus on the identification of emotion [74]. Therefore, the priming effects of two components would be affected by character of target task.

For another reason, it is worth mentioning that both the situational stimulus and target stimulus used in some earlier studies were pictures [42,43,50], whereas the situational stimulus in the current study was presented in the form of sentences. Research has indicated that people can extract emotional information from pictures quickly and effectively even if the picture is presented very briefly [67,75,76], which may reflect attentional capture driven by basic emotional stimulation. In some studies, when the situational stimulus is complex, such as abstract speech information, the priming effect often occurs in the later stage [56,70]. Therefore, the null results with respect to P2 may have occurred probably because the primes were in the form of sentences rather than pictures or words.

## Limitations

Several limitations should be noted. First, in the pretest we evaluated the sentence materials with the method used in Yang's studies [26,32]. That is, participants rated how morally wrong and how disgusting each sentence was. However, in some studies, participants reported that they stated they felt disgust because that was the only response option available, when they may actually have experienced a different emotion [70,71]. Thus, we cannot be sure that the level of disgust induced by the sentences was assessed objectively. Second, in the face searching task, the only negative face was disgust, and other negative emotions such as anger were not examined. Therefore, it is unknown whether participants' responses were specific to disgust or simply responses to negative stimuli. Future studies could use different negative emotions to compare the influence of moral violations with and without impurity on the processing of other types of emotional faces. Third, this is the first study to use a face searching task to study the effects of emotional primes. Therefore, P2, N400, and LPP could be impacted by the nature of the task and more research is needed to verify the priming effect in this study.

## Supporting information

**S1 Appendix.**
(DOCX)

**S1 File.**
(ZIP)

## Author Contributions

**Conceptualization:** Ming Peng.

**Data curation:** Siyu Jiang, Xiaohui Wang.

**Formal analysis:** Siyu Jiang.

**Funding acquisition:** Ming Peng.

**Investigation:** Siyu Jiang, Xiaohui Wang.

**Methodology:** Ming Peng.

**Project administration:** Ming Peng.

**Supervision:** Ming Peng.

**Visualization:** Siyu Jiang.

**Writing – original draft:** Siyu Jiang.

**Writing – review & editing:** Siyu Jiang, Ming Peng.

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
