## [Decision Letter · Decision Letter 0]

12 Aug 2020

PONE-D-20-17995

Different influences of moral violation with and without physical impurity on face processing: An Event-Related Potentials Study

PLOS ONE

Dear Dr. Peng,

Thank you for submitting your manuscript to PLOS ONE. After careful consideration, we feel that it has merit but does not fully meet PLOS ONE’s publication criteria as it currently stands. Therefore, we invite you to submit a revised version of the manuscript that addresses the points raised during the review process.

Three experts have reviewed your manuscript and provided detailed feedback. Although the reviewers see potential in the paper, they each raise significant concerns regarding the framework and rationale of the study, the clarity of the hypotheses and procedures, the analysis of the data, and the conclusions. I have read your paper independently and concur with the concerns raised by the reviewers. In a revision, all of the reviewers' points need to be addressed. Also, language regarding the extent to which these results demonstrate responses specific to disgust should be tempered. As trials involving faces expressing other negative emotions were not included, it is unknown whether findings are specific to disgust or simply negative stimuli. Similarly, although the sentences were pretested for self-reported disgust responses, other emotions were not assessed (e.g., anger, contempt), so it is unknown whether disgust is the only emotion that distinguishes the stimuli. Furthermore, there are examples in the literature of participants reporting that they feel disgust because that is the response option provided to them, when they may actually be experiencing a different emotions. These limitations, as well as other study limitations, should be addressed in the discussion. Please report means and standard deviations for the results from the stimuli pretesting. Also, please pay close attention to reporting of the results. There are a number of typos (e.g., line 161: MN and MD should be greater than NN, not less than; lines 237-238: NN<MN, p=.32, if nonsignificant, NN should be equal to MN).    

We look forward to receiving your revised manuscript.

Kind regards,

Natalie J. Shook

Academic Editor

PLOS ONE

Journal Requirements:

2. Please provide additional information about the participant recruitment method and the demographic details of your participants, such as a) a description of any inclusion/exclusion criteria that were applied to participant recruitment, b) a table of relevant demographic details, c) a statement as to whether your sample can be considered representative of a larger population, d) a description of how participants were recruited, and e) descriptions of where participants were recruited and where the research took place.

Reviewers' comments:

Reviewer's Responses to Questions

**Comments to the Author**

1. Is the manuscript technically sound, and do the data support the conclusions?

Reviewer #1: Partly

Reviewer #2: Yes

Reviewer #3: Partly

2. Has the statistical analysis been performed appropriately and rigorously? 

Reviewer #1: No

Reviewer #2: Yes

Reviewer #3: No

3. Have the authors made all data underlying the findings in their manuscript fully available?

Reviewer #1: No

Reviewer #2: Yes

Reviewer #3: Yes

4. Is the manuscript presented in an intelligible fashion and written in standard English?

Reviewer #1: Yes

Reviewer #2: Yes

Reviewer #3: Yes

5. Review Comments to the Author

Reviewer #1: In the study titled “Different influences of moral violation with and without physical impurity on face processing: An Event-Related Potentials Study”, the authors sought to answer the question of whether priming participants with moral violations that contained disgust-relevant content would influence a subsequent face-processing task differently than priming participants with moral violations that did not contain disgust-relevant content. The authors appear to hypothesize that there will be no difference between disgust-relevant and non-disgust-relevant primes with regard to their influence on face processing, although I found the authors’ specific predictions a bit difficult to follow, so I would suggest edits to ensure that the authors’ predictions are clearly presented at the outset of the manuscript.

The authors do a good job of summarizing prior research relevant to their study. The one thing that I feel would be relevant to the introduction and which would make the introduction stronger is a brief discussion of work that delineates the functional (i.e. disease avoidance) mechanism of disgust versus behaviors and psychological processes such as moral evaluation which are more likely to be by-products of evolved disgust processes (see ‘Moral Disgust and the Tribal Instincts Hypothesis’ [Kelley, 2013]). In other words, cite prior research arguing that although the emotion of disgust is likely to have evolved to promote disease avoidance, these processes were also likely to be useful and easily co-opted to promote avoidance of social transgressions. I do not think this is required for publication, but would strengthen the ‘no difference’ argument that the authors’ are setting forth in the introduction.

I feel that the research procedure needs a more clear explanation. Specifically, in the facial recognition part of the procedure, I believe that the authors presented participants with a pair of faces on every trial. On some trials, one of the faces depicted disgust and the other face was neutral, whereas on other trials, both faces were neutral. I would ask the authors to revise their description of the procedure and confirm whether my interpretation is correct, and if not, present a more clear and detailed description of this stage of the research procedure.

If the above interpretation of the face processing task is correct, it is important to point out that differences in reaction time between the two face conditions would be expected due to the fact that, on roughly half of the ‘disgust face’ trials (assuming random stimuli presentation), participants can make a correct judgment (i.e., ‘disgust present’) after only scanning one face, whereas in the ‘neutral face’ trials, participants always have to scan both faces in order to make a correct judgment. I do not have prior experience in working with ERP data, and so I’m not sure the implications that this would have for the ERP analysis.

I would strongly encourage the authors to add the results of a power/sensitivity analysis at the beginning of their results section. The sample size for their main study is small, and it would be important to know the number of additional participants that would be necessary in order for the findings to be nullified. I would also suggest that the authors at a minimum make a statement as to whether their reaction time and ERP measures were normally distributed, as violations of the normality assumption may undermine the research findings.

Ultimately, the researchers concluded from their results that priming moral violations with and without physical impurity both induce disgust as evidenced by faster facial recognition and differential ERP responses relative to neutral prime conditions. This conclusion is warranted. However, it does appear, at least from the reaction time studies, that priming participants with moral violations WITH physical impurity induced disgust to a significantly greater degree than the moral violations without physical impurity. This is evidenced by the significant differences between the MN and MD conditions in the expected direction in both the ‘disgust face’ (facilitation of recognition) and ‘neutral face’ (impedance of recognition). The authors should call attention to this at a minimum, as this appeared to run counter to their stated hypotheses.

Overall, I think the research makes a meaningful contribution and should be published with the revisions suggested above.

Reviewer #2: I think that the study seems to be well-designed. The statistical analyses are appropriate and presented well. I also believe that this study could make an important contribution to the literature. However, I do have concerns about the conceptual framework of the paper and believe that it needs significant rewriting before it is published. I recommend publication with revision.

1. The authors indicate that "There is an open question of whether disgust arises in response to moral violations without impurity." However, it is not clear that their procedure enables them address this question. They are observing the effect of moral violations on the processing speed of disgust faces. The authors seem to be conflating the processing of disgust faces with the elicitation of disgust. For example, in their concluding remarks, they make statements such as "moral transgression with physical impurity did induce disgust" (lines 396-397) and "moral violations without physical impurity also induced disgust" (lines 402-403). Unless the authors can make a convincing argument as to why the processing of disgust faces as an indication of induced disgust, they should reframe their introduction and concluding remarks to better align with their study.

2. The authors should include an appendix that contains all of the moral violation statements that were used in the study.

3. Figures should contain keys for acronyms.

4. Figure 7 was intended to show the topographical maps for both the neutral and disgust faces. However, the maps for the disgust faces appear to be omitted.

5. There seem to be numerous translational/grammatical issues throughout the document that need to be cleaned up.

Reviewer #3: The present paper used ERPs to examine whether the processing of disgust vs. neutral emotion faces differed depending on moral violation primes that contained or did not contain purity. ERP analyses suggest that moral violations played a role in the processing of disgust faces, but that this effect may be more prominent for moral violations with impurity. Although the topic of the paper is interesting, and uses more complex research techniques, the paper needs more theoretical foundation for the research, the analyses conducted are unclear and leave room for interpretation, and the data don’t always support claims made. Because of these reasons, I unfortunately can’t recommend the paper for publication in its current form. More detailed comments are below:

Introduction:

⁃ The Introduction reads very cursory, and could use more details in areas. Specifically, a better theoretical framework could be set up for the present research.

⁃ The authors should cite more work by Hanah Chapman and Joshua Tybur, who have conducted relevant research on disgust and purity, e.g, Karinen & Chapman, 2019; Giner-Sorolla & Chapman 2016.

⁃ The overall writing of the Introduction could be better structured (e.g., with subheadings) to make it easier to read/understand, especially by individuals unfamiliar with the topic.

⁃ The authors would benefit from making a more convincing case for why it is important to distinguish between moral violations with and without impurity. Again, a stronger theoretical framework would guide this.

⁃ Although the Introduction sets up the paper to emphasize the differences between moral violations with and without impurity, the hypotheses did not seem to distinguish between the two. This leads me to wonder why it is important to differentiate between these two types of moral violations.

Method:

⁃ How was the sample size determined? Through power analysis? Although the final sample of 25 participants is acceptable for ERP research, it is still somewhat low. So a better justification for the sample size would help.

⁃ The authors state that the wording of the sentences presented to participants were modified. How much modification of the wording occurred? Could the authors provide examples of these modifications?

⁃ The number of trials doesn’t seem to add up. The authors state that “Every block consisted of 112 trials. Each block consisted of three types of sentences, 42 sentences of each type.” If each block had all 42 sentences of each of the three sentence types, the number of trials per block should be 126, not 112. This could use some clarification.

⁃ The authors should provide more justification (based on previous work) for why they selected the electrodes they did for analyses.

Results:

⁃ The two N2pc analyses seem redundant. It would be sufficient to only run the analysis using difference scores, as that seems to be the norm for N2pc.

⁃ Also, the set up of the second N2pc analysis is unclear. Did that analysis examine face types? This wasn’t written in the paper, but the results did report on face types. The specifics of the analysis should be more clearly stated.

⁃ Although neutral face followed by MN sentences elicited larger N400 amplitudes, in Figure 6, it looks like the N400 is barely present across electrodes. Given that there was only one significant difference in N400 amplitudes across conditions, I worry about the reliability of the N400 result.

⁃ Furthermore, the N400 amplitudes appear to be all positive, which is a bit odd, given that it’s the N400 which is generally a negative-going waveform.

⁃ Generally, ERP reporting convention plots negative amplitudes above 0 and positive amplitudes below.

⁃ It may be worthwhile to examine the P600 or LPP instead of the P2, since faces are more complex stimuli, especially following reading complex sentences.

Discussion:

⁃ It’s not really appropriate to say that the current paradigm “induced” disgust, as participants weren’t asked to rate their own disgust levels, nor were measures taken directly from the participants about their emotional states. It is more accurate to say that the study is about how people “process” disgust, as they responded to existing disgusting or neutral stimuli.

⁃ On page 16 of the discussion, it says that “intensity was lower in the case of moral violation with physical impurity,” but page 17 states that “disgust in the case of moral violations with physical impurity was stronger.” These are contradictory statements, and are worthy of clarification.

⁃ The given data can’t conclude that the reason no P2 effects were found was due to stimulus type. I suggest wording that conclusion more carefully.

⁃ The discussion tends to overstate what the data has found. Quite a few of the claims made aren’t necessarily supported by the present data.

6. PLOS authors have the option to publish the peer review history of their article (what does this mean?). If published, this will include your full peer review and any attached files.

Reviewer #1: **Yes: **Russ Clay, PhD

Reviewer #2: No

Reviewer #3: **Yes: **Xiaowen Xu

---

## [Author Response · Author response to Decision Letter 0]

5 Oct 2020

List of Responses

EDITOR 

Thank you for submitting your manuscript to PLOS ONE. After careful consideration, we feel that it has merit but does not fully meet PLOS ONE’s publication criteria as it currently stands. Therefore, we invite you to submit a revised version of the manuscript that addresses the points raised during the review process.

Three experts have reviewed your manuscript and provided detailed feedback. Although the reviewers see potential in the paper, they each raise significant concerns regarding the framework and rationale of the study, the clarity of the hypotheses and procedures, the analysis of the data, and the conclusions. I have read your paper independently and concur with the concerns raised by the reviewers. In a revision, all of the reviewers' points need to be addressed. Also, language regarding the extent to which these results demonstrate responses specific to disgust should be tempered. As trials involving faces expressing other negative emotions were not included, it is unknown whether findings are specific to disgust or simply negative stimuli. Similarly, although the sentences were pretested for self-reported disgust responses, other emotions were not assessed (e.g., anger, contempt), so it is unknown whether disgust is the only emotion that distinguishes the stimuli. Furthermore, there are examples in the literature of participants reporting that they feel disgust because that is the response option provided to them, when they may actually be experiencing a different emotions. These limitations, as well as other study limitations, should be addressed in the discussion. Please report means and standard deviations for the results from the stimuli pretesting. Also, please pay close attention to reporting of the results. There are a number of typos (e.g., line 161: MN and MD should be greater than NN, not less than; lines 237-238: NN<MN, p=.32, if nonsignificant, NN should be equal to MN). 

Authors’ response:

Thank you very much for having offered us the opportunity to revise and resubmit our paper and for giving us constructive comments. We carefully examined all comments provided, and tried to adequately repair all issues addressed by the reviewers.

1. We agreed with the questions you raised, we illustrated this in the Limitation.

See:

Limitation:

“Several limitations should be noted. First, in the pretest we evaluated the sentence materials with the method used in Yang's studies [26,32]. That is, participants rated how morally wrong and how disgusting each sentence was. However, in some studies, participants reported that they stated they felt disgust because that was the only response option available, when they may actually have experienced a different emotion [70,71]. Thus, we cannot be sure that the level of disgust induced by the sentences was assessed objectively. Second, in the face searching task, the only negative face was disgust, and other negative emotions such as anger were not examined. Therefore, it is unknown whether participants’ responses were specific to disgust or simply responses to negative stimuli. Future studies could use different negative emotions to compare the influence of moral violations with and without impurity on the processing of other types of emotional faces. Third, this is the first study to use a face searching task to study the effects of emotional primes. Therefore, P2, N400, and LPP could be impacted by the nature of the task and more research is needed to verify the priming effect in this study.” 

2. We added means and standard deviations for the results in pretesting.

See line 212-227.

3. About line 210, when participants were asked to rate the degree to which each sentence was morally wrong, ratings were made on 1 = very immoral, 9 = very moral. That is, the smaller the score, the less immoral the behavior, so MN and MD were less than NN.

4. Lines 296-298: we have modified the typos.

See:

“ (F (2,54) = 10.28，p < 0.001; NN:857.469 ± 34.953 ms , WN:868.957 ± 35.306ms , WD:890.434 ± 37.915ms ; NN <WD, p< 0.001; WN<WD, p = 0.029; NN =WN, p = 0.32)”

 

REVIEWER #1

In the study titled “Different influences of moral violation with and without physical impurity on face processing: An Event-Related Potentials Study”, the authors sought to answer the question of whether priming participants with moral violations that contained disgust-relevant content would influence a subsequent face-processing task differently than priming participants with moral violations that did not contain disgust-relevant content. The authors appear to hypothesize that there will be no difference between disgust-relevant and non-disgust-relevant primes with regard to their influence on face processing, although I found the authors’ specific predictions a bit difficult to follow, so I would suggest edits to ensure that the authors’ predictions are clearly presented at the outset of the manuscript.

The authors do a good job of summarizing prior research relevant to their study. The one thing that I feel would be relevant to the introduction and which would make the introduction stronger is a brief discussion of work that delineates the functional (i.e. disease avoidance) mechanism of disgust versus behaviors and psychological processes such as moral evaluation which are more likely to be by-products of evolved disgust processes (see ‘Moral Disgust and the Tribal Instincts Hypothesis’ [Kelley, 2013]). In other words, cite prior research arguing that although the emotion of disgust is likely to have evolved to promote disease avoidance, these processes were also likely to be useful and easily co-opted to promote avoidance of social transgressions. I do not think this is required for publication, but would strengthen the ‘no difference’ argument that the authors’ are setting forth in the introduction.

Authors’ response:

We sincerely appreciate your positive feedback regarding our manuscript. Likewise, we fully appreciate the time and effort you put in this review, and the valuable suggestions to improve the suitability of the paper to the target audience.

We have added relevant contents to the introduction. At the beginning of the introduction, we first introduced the function of disgust and how disgust developed from avoidance of virus to avoidance of social behavior (part 1 in Introduction). Then we proposed that it is still controversial whether moral violation without impurity induced disgust (part 2 in Introduction).

I feel that the research procedure needs a more clear explanation. Specifically, in the facial recognition part of the procedure, I believe that the authors presented participants with a pair of faces on every trial. On some trials, one of the faces depicted disgust and the other face was neutral, whereas on other trials, both faces were neutral. I would ask the authors to revise their description of the procedure and confirm whether my interpretation is correct, and if not, present a more clear and detailed description of this stage of the research procedure.

Authors’ response:

Thank you for your suggestions. Your interpretation is correct. We have modified the description in the manuscript.

See: 

Method Line 241-242

“On half of the trials, one face described disgust, the other was neutral, and on the other half, both faces were neutral.”

At the same time, we have presented all types of face stimuli in the flow chart.

See:

If the above interpretation of the face processing task is correct, it is important to point out that differences in reaction time between the two face conditions would be expected due to the fact that, on roughly half of the ‘disgust face’ trials (assuming random stimuli presentation), participants can make a correct judgment (i.e., ‘disgust present’) after only scanning one face, whereas in the ‘neutral face’ trials, participants always have to scan both faces in order to make a correct judgment. I do not have prior experience in working with ERP data, and so I’m not sure the implications that this would have for the ERP analysis.

Authors’ response:

We are well aware of the fact that you tackled an important issue here, and we agree with your concern. In the search paradigm, there may be differences in the cognitive resources consumed by the subjects under the two conditions, and the processing of emotional faces will be different. Therefore, in order to avoid the possible differences between the two conditions, we did not directly compare the differences between disgust face condition and neutral face condition. We compared the neutral and the disgust face condition separately, that is, whether the priming effect of the three sentences is different under disgust face condition or neutral face condition.

I would strongly encourage the authors to add the results of a power/sensitivity analysis at the beginning of their results section. The sample size for their main study is small, and it would be important to know the number of additional participants that would be necessary in order for the findings to be nullified. I would also suggest that the authors at a minimum make a statement as to whether their reaction time and ERP measures were normally distributed, as violations of the normality assumption may undermine the research findings.

Authors’ response:

The sample size was comparable to those of previous ERP studies examining the moral processing [1, 2, 3].

According to your suggestion, the results of post-hoc test are as follows.

See:

“ Using this sample size (N = 25) and a pre-defined effect size (ηp2) of 0.25, a power analysis in G*Power [4] showed that this sample size would give over 90% power to detect an effect.”

Reference

1.Luo Y, Shen WL, Zhang Y, Feng TY, Huang H, Li H. Core disgust and m oral disgust are related to distinct spatiotem poral patterns of neural processing: An event-related potential study. Biological Psychology. 2013; 94:242-248. 

2. Peng XZ, Jiao C, Cui F, Chen QF, Li P, Li H. The time course of indirect moral judgment in gossip processing modulated by different agents. Psychophysiology. 2017; 54:1459-1471. 

3.Zhang XY, Guo Q, Zhang YX, Lou LD, Ding DQ. Different Timing Features in Brain Processing of Core and Moral Disgust Pictures: An Event-Related Potentials Study. PLoS One. 2015; 10(5): e0128531. 

4.Faul F, Erdfelder E, Lang AG, Buchner A. G~*power 3: a flexible statistical power analysis program for the social, behavioral, and biomedical sciences. Behavior Research Methods. 2007; 39(2), 175-191.

Ultimately, the researchers concluded from their results that priming moral violations with and without physical impurity both induce disgust as evidenced by faster facial recognition and differential ERP responses relative to neutral prime conditions. This conclusion is warranted. However, it does appear, at least from the reaction time studies, that priming participants with moral violations WITH physical impurity induced disgust to a significantly greater degree than the moral violations without physical impurity. This is evidenced by the significant differences between the MN and MD conditions in the expected direction in both the ‘disgust face’ (facilitation of recognition) and ‘neutral face’ (impedance of recognition). The authors should call attention to this at a minimum, as this appeared to run counter to their stated hypotheses.

Overall, I think the research makes a meaningful contribution and should be published with the revisions suggested above.

Authors’ response:

We agree that we should have provided more detailed information to interpret priming effect. We illustrated it in the Introduction and Discussion.

In the Introduction:

“In affective priming studies [42-45], participants are presented with background information (pictures, words, sentences), and then presented with faces showing different affective expressions. Facilitated processing (reaction times) is showed in affectively congruent targets and impaired processing is showed in incongruent ones, a phenomenon known as the affective priming effect. The presentation of priming stimuli can automatically activate the related emotional representations of the brain and carry out implicit emotional processing [46,47]. Multiple studies have shown the priming effect, and this effect can be also detected using ERP data [45,48,49]. ”

In the Discussion:

“In the affective priming task [42,45,69,70], facilitated processing was shownwhen the prime and target were affectively congruent, and impaired processing was shown when the prime and target were emotionally incongruent, that is the affective priming effect. Such an effect shows that the emotional information provided by a stimulus can be implicitly and automatically evaluated.”

As to behavioral results, we found that MN and MD facilitate the detection of disgust face, however, only MD impedance recognition of neutral face. Therefore, we infer that the influence of moral violations with physical impurity was stronger than moral violations without physical impurity.

Finally, we would like to say that we really appreciated your careful examination of our paper and the many valuable suggestions you made to improve our paper. We feel that we have learned a lot from the specific comments you made.

References:

42. Righart R, Gelder BD. Rapid influence of emotional scenes on encoding of facial expressions: an ERP study. Social Cognitive & Affective Neuroscience. 2008; 3(3): 270.

43. Lu Y, Zhang WN, Hu W, Luo YJ. Understanding the subliminal affective priming effect of facial stimuli: an ERP study. Neurosci Lett. 2011; 502(3): 182-185.

44. Li S, Li P, Wang W, Zhu X, Luo W. The effect of emotionally valenced eye region images on visuocortical processing of surprised faces. Psychophysiology. 2017; 55(1): 12.

45. Fazio, RH. On the automatic activation of associated evaluations: An overview. Cognition and Emotion. 2001; 15(2): 115-141.

46. Kobylinska D, Karwowska D. Assimilation and contrast effects in suboptimal affective priming paradigm. Frontiers in Psychology. 2014; 5.

47. Gibbons H, Seib-Pfeifer L, Koppehele-Gossel J, Schnuerch R. Affective priming and cognitive load: Event-related potentials suggest an interplay of implicit affect misattribution and strategic inhibition. Psychophysiology. 2017; 55(4): e13009.

48. Aguado L, Dieguez-Risco T, Méndez-Bértolo C, Pozo MA, Hinojosa JA. Priming effects on the N400 in the affective priming paradigm with facial expressions of emotion. Cognitive, Affective, & Behavioral Neuroscience. 2013; 13(2): 284-296.

49. Werheid K, Alpay G, Jentzsch I, Sommer W. Priming emotional facial expressions as evidenced by event-related brain potentials. International Journal of Psychophysiology. 2005; 55(2): 209-219.

69. Klauer KC, Musch J. Affective priming: Findings and theories. In: Musch J, Klauer KC, editors. The Psychology of Evaluation: Affective Processes in Cognition and Emotion. Lawrence Erlbaum: 2003. pp. 7-49.

70. Diéguez-Risco T, Aguado L, Albert J, Hinojosa JA. Faces in context: Modulation of expression processing by situational information. Social Neuroscience. 2013; 8(6): 601-620.

 

REVIEWER #2

I think that the study seems to be well-designed. The statistical analyses are appropriate and presented well. I also believe that this study could make an important contribution to the literature. However, I do have concerns about the conceptual framework of the paper and believe that it needs significant rewriting before it is published. I recommend publication with revision.

Authors’ response:

We were pleased to hear that you appreciated our study. Thank you for your constructive comments and your helpful suggestions to improve the suitability of the paper.

The authors indicate that "There is an open question of whether disgust arises in response to moral violations without impurity." However, it is not clear that their procedure enables them address this question. They are observing the effect of moral violations on the processing speed of disgust faces. The authors seem to be conflating the processing of disgust faces with the elicitation of disgust. For example, in their concluding remarks, they make statements such as "moral transgression with physical impurity did induce disgust" (lines 396-397) and "moral violations without physical impurity also induced disgust" (lines 402-403). Unless the authors can make a convincing argument as to why the processing of disgust faces as an indication of induced disgust, they should reframe their introduction and concluding remarks to better align with their study.

Authors’ response:

We are very grateful to the reviewer for pointing out the problems in the article. In the new version, we introduce the mechanism of paradigm more specifically. In the beginning of Discussion, we describe the results more objectively and are cautious to infer the implication of results.

See:

In the Introduction:

“In affective priming studies [42-45], participants are presented with background information (pictures, words, sentences), and then presented with faces showing different affective expressions. Facilitated processing (reaction times) is showed in affectively congruent targets and impaired processing is showed in incongruent ones, a phenomenon known as the affective priming effect. The presentation of priming stimuli can automatically activate the related emotional representations of the brain and carry out implicit emotional processing [46,47]. Multiple studies have shown the priming effect, and this effect can be also detected using ERP data [45,48,49]. ”

In the Discussion:

“In the affective priming task [42,45,69,70], facilitated processing was shown when the prime and target were affectively congruent, and impaired processing was shown when the prime and target were emotionally incongruent, that is the affective priming effect. Such an effect shows that the emotional information provided by a stimulus can be implicitly and automatically evaluated.”

As to behavioral results, we found that MN and MD facilitate the detection of disgust face, however, only MD impedance recognition of neutral face. Therefore, we infer that the influence of moral violations with physical impurity was stronger than moral violations without physical impurity.

What’s more, we have revised the abstract and further combed the theoretical framework of the introduction.

References:

42. Righart R, Gelder BD. Rapid influence of emotional scenes on encoding of facial expressions: an ERP study. Social Cognitive & Affective Neuroscience. 2008; 3(3): 270.

43. Lu Y, Zhang WN, Hu W, Luo YJ. Understanding the subliminal affective priming effect of facial stimuli: an ERP study. Neurosci Lett. 2011; 502(3): 182-185.

44. Li S, Li P, Wang W, Zhu X, Luo W. The effect of emotionally valenced eye region images on visuocortical processing of surprised faces. Psychophysiology. 2017; 55(1): 12.

45. Fazio, RH. On the automatic activation of associated evaluations: An overview. Cognition and Emotion. 2001; 15(2): 115-141.

46. Kobylinska D, Karwowska D. Assimilation and contrast effects in suboptimal affective priming paradigm. Frontiers in Psychology. 2014; 5.

47. Gibbons H, Seib-Pfeifer L, Koppehele-Gossel J, Schnuerch R. Affective priming and cognitive load: Event-related potentials suggest an interplay of implicit affect misattribution and strategic inhibition. Psychophysiology. 2017; 55(4): e13009.

48. Aguado L, Dieguez-Risco T, Méndez-Bértolo C, Pozo MA, Hinojosa JA. Priming effects on the N400 in the affective priming paradigm with facial expressions of emotion. Cognitive, Affective, & Behavioral Neuroscience. 2013; 13(2): 284-296.

49. Werheid K, Alpay G, Jentzsch I, Sommer W. Priming emotional facial expressions as evidenced by event-related brain potentials. International Journal of Psychophysiology. 2005; 55(2): 209-219.

69. Klauer KC, Musch J. Affective priming: Findings and theories. In: Musch J, Klauer KC, editors. The Psychology of Evaluation: Affective Processes in Cognition and Emotion. Lawrence Erlbaum: 2003. pp. 7-49.

70. Diéguez-Risco T, Aguado L, Albert J, Hinojosa JA. Faces in context: Modulation of expression processing by situational information. Social Neuroscience. 2013; 8(6): 601-620.

The authors should include an appendix that contains all of the moral violation statements that were used in the study.

Authors’ response:

According to your opinion, we have added the appendix of experimental materials.

Figures should contain keys for acronyms.

Authors’ response:

Thanks for your comment, we have annotated the abbreviations.

Figure 7 was intended to show the topographical maps for both the neutral and disgust faces. However, the maps for the disgust faces appear to be omitted.

Authors’ response:

We are very grateful to the reviewer for pointing out the problem. We have added the maps for the disgust faces.

See

Fig. 7. Topographical maps of voltage amplitudes of N400 in response to disgust face and neutral face across the NN, WN, and WD priming conditions.

There seem to be numerous transitional/grammatical issues throughout the document that need to be cleaned up.

Authors’ response:

As you pointed out correctly, the author team does not include a single native English speaker. In order to reduce the transitional/grammatical errors, we have asked a native proofreader to make an overall revision of the article.

 

REVIEWER #3

The present paper used ERPs to examine whether the processing of disgust vs. neutral emotion faces differed depending on moral violation primes that contained or did not contain purity. ERP analyses suggest that moral violations played a role in the processing of disgust faces, but that this effect may be more prominent for moral violations with impurity. Although the topic of the paper is interesting, and uses more complex research techniques, the paper needs more theoretical foundation for the research, the analyses conducted are unclear and leave room for interpretation, and the data don’t always support claims made. Because of these reasons, I unfortunately can’t recommend the paper for publication in its current form. More detailed comments are below:

Introduction:

⁃ The Introduction reads very cursory, and could use more details in areas. Specifically, a better theoretical framework could be set up for the present research.

Authors’ response:

We are very grateful to the reviewer for giving us valuable suggestions.

Following up on this recommendation, we have substantially revised the introduction and extended the literature on which it is based. The revised version is meant to provide a clearer theoretical framework. 

⁃ The authors should cite more work by Hanah Chapman and Joshua Tybur, who have conducted relevant research on disgust and purity, e.g, Karinen & Chapman, 2019; Giner-Sorolla & Chapman 2016.

Authors’ response:

Thanks for your kind suggestion to help us improve our Introduction. We rewrote the Introduction and added some important works that you mentioned.

In the first part of Introduction, we introduced the evolutionary origins of disgust and morality. In the second part of Introduction, two conflicting views and its evidence on the relationship between disgust and morality was introduced.

See part 2 in Introduction 

Line 46-73.

References:

Karinen, A. K., & Chapman, H. A. (2019). Cognitive and personality correlates of trait disgust and their relationship to condemnation of non-purity moral transgressions. Emotion, 19(5), 889–902.

Giner-Sorolla, R., & Chapman, H. A. (2017). Beyond Purity: Moral Disgust Toward Bad Character. Psychological Science, 28(1), 80-91.

⁃ The overall writing of the Introduction could be better structured (e.g., with subheadings) to make it easier to read/understand, especially by individuals unfamiliar with the topic.

Authors’ response:

We are very grateful to the reviewer for pointing out the problem and giving us very useful suggestion. We added subheadings to make each part of the article clearer and more readable. 

⁃ The authors would benefit from making a more convincing case for why it is important to distinguish between moral violations with and without impurity. Again, a stronger theoretical framework would guide this.

Authors’ response:

We agree that, in our original paper, there was room for a better alignment of the different parts of the Introduction. Your remark made us realize that we had to systematically work on the integration of all parts. Hoping the new version meets your expectations and publication standards. 

⁃ Although the Introduction sets up the paper to emphasize the differences between moral violations with and without impurity, the hypotheses did not seem to distinguish between the two. This leads me to wonder why it is important to differentiate between these two types of moral violations.

Authors’ response:

In previous articles, the statement is not clear enough. Our purpose is not to emphasize the difference between moral violations with and without impurity, but to emphasize the uncertainty of the relationship between the two. In the new version, in the Introduction, we objectively introduce two kinds of contradictory viewpoints and existing evidence.

Based on the results of a series of studies, we hypothesize that both moral violations with and without physical impurity will influence disgust face processing, however, their influence on face processing will have different temporal character on neural activity.

Method:

⁃ How was the sample size determined? Through power analysis? Although the final sample of 25 participants is acceptable for ERP research, it is still somewhat low. So a better justification for the sample size would help.

Authors’ response:

The sample size was comparable to those of previous ERP studies examining the moral processing [1, 2, 3].

According to your suggestion, the results of post-hoc test are as follows.

See:

“Using this sample size (N = 25) and a pre-defined effect size (ηp2) of 0.25, a power analysis in G*Power [4] showed that this sample size would give over 90% power to detect an effect.”

Reference

1.Luo Y, Shen WL, Zhang Y, Feng TY, Huang H, Li H. Core disgust and m oral disgust are related to distinct spatiotem poral patterns of neural processing: An event-related potential study. Biological Psychology. 2013; 94:242-248. 

2. Peng XZ, Jiao C, Cui F, Chen QF, Li P, Li H. The time course of indirect moral judgment in gossip processing modulated by different agents. Psychophysiology. 2017; 54:1459-1471. 

3.Zhang XY, Guo Q, Zhang YX, Lou LD, Ding DQ. Different Timing Features in Brain Processing of Core and Moral Disgust Pictures: An Event-Related Potentials Study. PLoS One. 2015; 10(5): e0128531. 

4.Faul F, Erdfelder E, Lang AG, Buchner A. G~*power 3: a flexible statistical power analysis program for the social, behavioral, and biomedical sciences. Behavior Research Methods. 2007; 39(2), 175-191.

⁃ The authors state that the wording of the sentences presented to participants were modified. How much modification of the wording occurred? Could the authors provide examples of these modifications?

Authors’ response:

For example, the sentence of the previous experimental materials was "At a public swimming pool, a person is shitting", and the revised sentence was "At a public swimming pool, a person is shitting", which is more in line with our usual language habits after adjustment.

⁃ The number of trials doesn’t seem to add up. The authors state that “Every block consisted of 112 trials. Each block consisted of three types of sentences, 42 sentences of each type.” If each block had all 42 sentences of each of the three sentence types, the number of trials per block should be 126, not 112. This could use some clarification.

Authors’ response:

Thank you very much for pointing out this problem.

There were 224 sentences for each sentence type and 672 trials in the whole experiment. The experiment was divided into six blocks, 112 trials for each experimental block. 

There were three types of sentences, and each type had 42 sentences. For one type, each sentence was repeated five times, then acquired 42×5=210 sentences. 14 sentences in each type, which were selected randomly, were repeated one more time. Therefore, 224 sentences were presented in each type.

672 trials= NN (224trials=42sentences×5times+14sentences) + MN(224trials) + MD(224trials)

⁃ The authors should provide more justification (based on previous work) for why they selected the electrodes they did for analyses.

Authors’ response:

Thanks for your comments, we added temporal and brain regions of P2, N400, and N2pc in previous studies in the Introduction. Present study is based on these studies and character of data in the study.

See:

“P2,which usually peaks around 200-250 ms, is located over the centro-frontal and the parieto-occipital region. It represents some aspects of higher-order perceptual processing, modulated by attention to visual stimuli[52]. N400 is a negative deflection observed around 400 ms after target presentation at centro-parietal scalp electrodes. It is sensitive to semantic relatedness and congruency [48].

N2pc was also observed over occipital scalp electrodes in the time range of 180-300 ms after stimulus onset contralateral to the side of an attended visual event [60].”

Reference

48. Aguado L, Dieguez-Risco T, Méndez-Bértolo C, Pozo MA, Hinojosa JA. Priming effects on the N400 in the affective priming paradigm with facial expressions of emotion. Cognitive, Affective, & Behavioral Neuroscience. 2013; 13(2): 284-296.

52. Meaux E, Hernandez N, Carteau-Martin I, Martineau J, Barthélémy C, Bonnet-Brilhault F, Batty M. Event-related potential and eye tracking evidence of the developmental dynamics of face processing. European Journal of Neuroscience. 2014; 39(8): 1349-1362.

60. Eimer M, Kiss M. Attentional capture by task-irrelevant fearful faces is revealed by the N2pc component. Biological Psychology. 2007; 74(1): 108-112.

Results:

⁃ The two N2pc analyses seem redundant. It would be sufficient to only run the analysis using difference scores, as that seems to be the norm for N2pc.

Authors’ response:

Thanks for your suggestion; we have deleted first result of N2pc.

⁃ Also, the set up of the second N2pc analysis is unclear. Did that analysis examine face types? This wasn’t written in the paper, but the results did report on face types. The specifics of the analysis should be more clearly stated.

Authors’ response:

The analysis of N2pc did not include the face type as a variable becauseN2pc was the difference waves between average waveform of disgust faces at the contralateral and ipsilateral electrodes. In order to reduce the mistakes in writing, we checked the results again.

See

“We conducted repeated measures ANOVAs on N2pc different waves (the contralateral waveform minus the ipsilateral waveform) with sentence types as a within-subject factor. The ANOVA results showed that the main effect of sentence types was significant (F (2,48) = 7.032, p = 0.002, ηp2= 0.227), under the NN priming condition, disgust face elicited larger amplitudes, compared with the WD priming condition (-1.114 vs. -0.708, p= 0.005) and the WN priming condition (-1.064 vs. -0.806, p= 0.072). No significant difference was observed between the WN priming condition and the WD priming condition (-0.806 vs. -0.708, p= 0.904).”

⁃ Although neutral face followed by MN sentences elicited larger N400 amplitudes, in Figure 6, it looks like the N400 is barely present across electrodes. Given that there was only one significant difference in N400 amplitudes across conditions, I worry about the reliability of the N400 result.

Authors’ response:

N400 is a classical component in the priming paradigm; greater amplitude was induced under inconsistent conditions. In previous studies, the amplitude of N400 component is different due to material tasks and other factors. For example, the N400 components in some studies are presented as follows.

See:

Reference:

Aguado L, Dieguez-Risco T, Méndez-Bértolo C, Pozo MA, Hinojosa JA. Priming effects on the N400 in the affective priming paradigm with facial expressions of emotion. Cognitive, Affective, & Behavioral Neuroscience. 2013; 13(2): 284-296

Reference:

Hietanen JK, Astikainen P. N170 response to facial expressions is modulated by the affective congruency between the emotional expression and preceding affective picture. Biological Psychology. 2013; 92(2):114-124.

The target task used in the study could be a factor that influenced N400. In previous studies, the target task was judgment of single-face or single-word, instead of the search task used in our study. Therefore, the amplitude of N400 component could be also influenced by the nature of task. It needs to be verified by more studies. In order to draw the reader's attention to this problem, we explain it in the Discussion and Limitations.

See：

Discussion:

“One of the reasons for the difference we inferred is different target task used in this study. When the priming task was used in previous studies, the target task was judgment of a single face or a single word—that is, a recognition task—instead of the search task used in our study [48,56]. Whereas face searching tasks emphasize the attentional orientation or how attention was captured by emotion [73], recognition tasks focus on the identification of emotion [74]. Therefore, the priming effects of two components would be affected by character of target task.”

Limitations:

“Third, this is the first study to usea face searching task to study the effects of emotional primes. Therefore, P2, N400, and LPP could be impacted by the nature of the task and more research is needed to verify the priming effect in this study.”

References:

48. Aguado L, Dieguez-Risco T, Méndez-Bértolo C, Pozo MA, Hinojosa JA. Priming effects on the N400 in the affective priming paradigm with facial expressions of emotion. Cognitive, Affective, & Behavioral Neuroscience. 2013; 13(2): 284-296.

56. Klein F, Iffland B, Schindler S, Wabnitz P, Neuner F. This person is saying bad things about you: The influence of physically and socially threatening context information on the processing of inherently neutral faces. Cognitive, Affective, & Behavioral Neuroscience. 2015; 15(4): 736-748.

73. Ottmar VL, Belinda MC, Mareka JF, Deborah JT,Joanne RS. Searching for emotion or race: Task-irrelevant facial cues have asymmetrical effects. Cognition and Emotion. 2014; 28(6): 1100-1109.

74. Smith FW, Rossit S. Identifying and detecting facial expressions of emotion in peripheral vision. PLoS ONE. 2018; 13(5): e0197160.

⁃ Furthermore, the N400 amplitudes appear to be all positive, which is a bit odd, given that it’s the N400 which is generally a negative-going waveform.

Authors’ response:

The N400 is a negative deflection appearing around 400 ms after stimulus onset. N400 in our study is also negative deflection. 

The graphs in the original version did not represent the waveform of each component well, so we marked out each component in the waveform.

See：

Fig 6. Grand average ERP waveforms recorded from C1, Cz, C2, CP1, CPz and CP2 in response to disgust face and neutral face across the NN, WN, and WD priming conditions.

⁃ Generally, ERP reporting convention plots negative amplitudes above 0 and positive amplitudes below.

Authors’ response:

Thanks for the suggestion. All figures of ERP waveform have been adjusted. 

⁃ It may be worthwhile to examine the P600 or LPP instead of the P2, since faces are more complex stimuli, especially following reading complex sentences.

Authors’ response:

Thanks for your suggestion; LPP is also an important component in affective priming task. Therefore, we added LPP in our manuscript. 

See:

Introduction 

“LPP is a central-parietal, midline component that becomes evident after 300 ms following the presentation of emotional stimuli onset and can be increased for several seconds [50]. It has been associated with emotional processing of faces and appraisal of affective meaning [51,52]. ”

Results 

“We conducted 3 (sentence types: NN, WN, WD) by 2 (facial types: disgust face, neutral face) repeated measures ANOVA for the LPP. The ANOVA results showed that the main effect of sentence types was not significant (F (2,48) = 0.404, p = 0.670, ηp² = 0.017), the main effect of facial types was not significant (F (1,24) = 0.048, p = 0.828, ηp² = 0.002), the interaction between sentence typesand facial types was significant (F (2,48) = 3.736, p = 0.031, ηp² = 0.135). Additional simple effects analyses revealed that sentence type effect was significant in the neutral face condition (F (2,48) = 3.80, p = 0.029). Post event multiple comparative analysis found that there was no significant difference between the sentence types under the disgust face condition (NN = WN, p = 0.089; NN = WD, p = 1.000; WN = WD, p= 0.158). There was no sentence type effect in the disgust face condition (p = 0.469).”

Discussion

“However, P2 and LPP results were inconsistent with our hypothesis. Wefound no significant difference in P2 and LPP amplitudes induced by neutral faces and disgusted faces under different types of moral violation priming conditions. These results suggest that face processing was not modulated by different types of moral violation. This finding is inconsistent with what would be expected based on some affective priming studies that found that priming effect would be observed on P2 and LPP amplitudes [46,47,53]. ”

References:

46. Kobylinska D, Karwowska D. Assimilation and contrast effects in suboptimal affective priming paradigm. Frontiers in Psychology. 2014; 5.

47. Gibbons H, Seib-Pfeifer L, Koppehele-Gossel J, Schnuerch R. Affective priming and cognitive load: Event-related potentials suggest an interplay of implicit affect misattribution and strategic inhibition. Psychophysiology. 2017; 55(4): e13009.

50. Hirai M, Watanabe S, Honda Y, Miki K, Kakigi R. Emotional object and scene stimuli modulate subsequent face processing: an event-related potential study. Brain Research Bulletin. 2008; 77(5): 264-273.

51. Krombholz A, Schaefer F, Boucsein W. Modification of N170 by different emotional expression of schematic faces. Biological Psychology. 2007; 76(3): 156-162.

52. Meaux E, Hernandez N, Carteau-Martin I, Martineau J, Barthélémy C, Bonnet-Brilhault F, Batty M. Event-related potential and eye tracking evidence of the developmental dynamics of face processing. European Journal of Neuroscience. 2014; 39(8): 1349-1362.

53. Hajcak G, Dunning JP, Foti D. Motivated and controlled attention to emotion: Time-course of the late positive potential. Clinical Neurophysiology: Official Journal of the International Federation of Clinical Neurophysiology. 2009; 120(3): 505-510.

Discussion:

⁃ It’s not really appropriate to say that the current paradigm “induced” disgust, as participants weren’t asked to rate their own disgust levels, nor were measures taken directly from the participants about their emotional states. It is more accurate to say that the study is about how people “process” disgust, as they responded to existing disgusting or neutral stimuli.

Authors’ response:

Thank you very much for pointing out this problem and giving us kind suggestions. 

Because subjective praise on emotion induced by moral violation behaviors is often influenced by many factors, such as semantic confusion of some emotions, we used affective priming paradigm to measure the implicit emotion induced by moral violation behaviors. We introduce the mechanism of paradigm more specifically in the Introduction.

See

“In affective priming studies [42-45], participants are presented with background information (pictures, words, sentences), and then presented with faces showing different affective expressions. Facilitated processing (reaction times) is showed in affectively congruent targets and impaired processing is showed in incongruent ones, a phenomenon known as the affective priming effect. The presentation of priming stimuli can automatically activate the related emotional representations of the brain and carry out implicit emotional processing [46,47].” 

From our behavioral result:

WD:

Participants' reaction time in detecting the disgusted faces under the priming condition of WD sentences was faster than under the priming condition of NN sentences; participants' reaction time in detecting neutral faces under the priming condition of WD sentences was slower than under the priming condition of NN sentences. These results suggested that WD, which facilitated the recognition of the disgusted face and hindered the recognition of the neutral face, therefore, we inferred WD would elicit disgust.

WN:

Participants' reaction time in recognizing disgusted faces under the priming condition of WN sentences was faster than under the priming condition of NN sentences. We inferred from this result that moral violation behaviors induce disgust.

However, the questions raised by you are also very worthy of attention. Therefore,we described “affective priming effect” in the Introduction and Discussion to help readers understand.

References:

42. Righart R, Gelder BD. Rapid influence of emotional scenes on encoding of facial expressions: an ERP study. Social Cognitive & Affective Neuroscience. 2008; 3(3): 270.

43. Lu Y, Zhang WN, Hu W, Luo YJ. Understanding the subliminal affective priming effect of facial stimuli: an ERP study. Neurosci Lett. 2011; 502(3): 182-185.

44. Li S, Li P, Wang W, Zhu X, Luo W. The effect of emotionally valenced eye region images on visuocortical processing of surprised faces. Psychophysiology. 2017; 55(1): 12.

45. Fazio, RH. On the automatic activation of associated evaluations: An overview. Cognition and Emotion. 2001; 15(2): 115-141.

46. Kobylinska D, Karwowska D. Assimilation and contrast effects in suboptimal affective priming paradigm. Frontiers in Psychology. 2014; 5.

47. Gibbons H, Seib-Pfeifer L, Koppehele-Gossel J, Schnuerch R. Affective priming and cognitive load: Event-related potentials suggest an interplay of implicit affect misattribution and strategic inhibition. Psychophysiology. 2017; 55(4): e13009.

⁃ On page 16 of the discussion, it says that “intensity was lower in the case of moral violation with physical impurity,” but page 17 states that “disgust in the case of moral violations with physical impurity was stronger.” These are contradictory statements, and are worthy of clarification.

Authors’ response:

We are appreciate that you have spotted an error (more specifically: an inconsistency) in our paper text. We have rewritten the sentence and checked all the content.

⁃ The given data can’t conclude that the reason no P2 effects were found was due to stimulus type. I suggest wording that conclusion more carefully.

Authors’ response:

Thanks for your kind suggestion. 

After comparing the differences among the affective priming experiments, we found that the type of priming stimulus could affect the processing of the target face.

Research has indicated that people can extract emotional information from pictures quickly and effectively even if the picture is presented very briefly [60-62], which may reflect attentional capture driven by basic emotional stimulation. In some studies, when the situational stimulus is complex, such as abstract speech information, the priming effect often occurs in the later stage [53,67]. Therefore, the null results with respect to P2 may have occurred probably because the primes were in the form of sentences rather than pictures or words. 

However, stimuli type might not be the only reason which led to this effect. Therefore, we also inferred that task character would also influence P2 effects. In the Discussion, we added this on.

See:

“One of the reasons for the difference we inferred is different target task used in this study. When the priming task was used in previous studies, the target task was judgment of a single face or a single word—that is, a recognition task—instead of the search task used in our study [45,53]. Whereas face searching tasks emphasize the attentional orientation or how attention was captured by emotion [70], recognition tasks focus on the identification of emotion [71]. Therefore, the priming effects of two components would be affected by character of target task.”

References:

45. Fazio, RH. On the automatic activation of associated evaluations: An overview. Cognition and Emotion. 2001; 15(2): 115-141.

53. Hajcak G, Dunning JP, Foti D. Motivated and controlled attention to emotion: Time-course of the late positive potential. Clinical Neurophysiology: Official Journal of the International Federation of Clinical Neurophysiology. 2009; 120(3): 505-510.

67. Borg JS, Lieberman D, Kiehl KA. Infection, incest, and iniquity: Investigating the neural correlates of disgust and morality. Journal of Cognitive Neuroscience. 2008; 20(9): 1529-1546.

70. Diéguez-Risco T, Aguado L, Albert J, Hinojosa JA. Faces in context: Modulation of expression processing by situational information. Social Neuroscience. 2013; 8(6): 601-620.

71. Galfano G, Sarlo M, Sassi F, Munafò M, Fuentes LJ, Umiltà C. Reorienting of spatial attention in gaze cuing is reflected in N2pc. Social Neuroscience. 2011; 6(3): 257-269.

⁃ The discussion tends to overstate what the data has found. Quite a few of the claims made aren’t necessarily supported by the present data.

Authors’ response:

We really appreciated your careful examination of our paper and the many valuable suggestions you made to improve our paper. In the new version, we describe the results more objectively and are cautious to infer the implication of results.

---

## [Decision Letter · Decision Letter 1]

12 Nov 2020

PONE-D-20-17995R1

Different influences of moral violation with and without physical impurity on face processing: An Event-Related Potentials Study

PLOS ONE

Dear Dr. Peng,

Thank you for submitting your manuscript to PLOS ONE. After careful consideration, we feel that it has merit but does not fully meet PLOS ONE’s publication criteria as it currently stands. Therefore, we invite you to submit a revised version of the manuscript that addresses the points raised during the review process.

First, I greatly appreciate your detailed attention to the reviewers' initial comments and suggestions. The manuscript is much stronger. However, the interpretation of your results and the language used to explain your results need to be tempered. Given the design of the pilot study and the main study, it cannot be concluded that the moral violation sentences (with or without impurity) elicited or induced disgust. As you note in the limitations section, other emotions or alternative explanations cannot be ruled out. The description of your findings needs to be precise (e.g., moral violation sentences facilitated processing of disgust faces). Similarly, please make sure that the description of previous work in the introduction is precise. For example, around line 85, you conclude that moral violations activate disgust because of overlapping neural activity from disgust and moral violation stimuli. Despite the similar patterns of neural activity, we cannot conclude that disgust was activated, rather all we can say is that neural activity associated with disgust was observed with the presentation of moral violation stimuli. Other, different neural activity patterns were also observed. Please revise the manuscript tightening up the language throughout to more precisely describe results.   

We look forward to receiving your revised manuscript.

Kind regards,

Natalie J. Shook

Academic Editor

PLOS ONE

Reviewers' comments:

Reviewer's Responses to Questions

**Comments to the Author**

1. If the authors have adequately addressed your comments raised in a previous round of review and you feel that this manuscript is now acceptable for publication, you may indicate that here to bypass the “Comments to the Author” section, enter your conflict of interest statement in the “Confidential to Editor” section, and submit your "Accept" recommendation.

Reviewer #1: All comments have been addressed

Reviewer #2: All comments have been addressed

2. Is the manuscript technically sound, and do the data support the conclusions?

Reviewer #1: Yes

Reviewer #2: Yes

3. Has the statistical analysis been performed appropriately and rigorously? 

Reviewer #1: Yes

Reviewer #2: Yes

4. Have the authors made all data underlying the findings in their manuscript fully available?

Reviewer #1: No

Reviewer #2: Yes

5. Is the manuscript presented in an intelligible fashion and written in standard English?

Reviewer #1: Yes

Reviewer #2: Yes

6. Review Comments to the Author

Reviewer #1: The authors have adequately addressed the feedback I provided in my initial review. I feel that the manuscript makes a valuable contribution to the literature and is acceptable for publication pending approval from additional reviewers and the editor. It appears as though the authors are not making their data set publicly available. If it is at all possible, I would recommend that the authors make all data associated with the study available for public use, specifically for replication research.

Reviewer #2: The authors have satisfactorily addressed all of the concerns that I had with the original manuscript. I recommend publication.

7. PLOS authors have the option to publish the peer review history of their article (what does this mean?). If published, this will include your full peer review and any attached files.

Reviewer #1: **Yes: **Russ Clay, PhD

Reviewer #2: No

---

## [Author Response · Author response to Decision Letter 1]

19 Nov 2020

Thank you very much for giving us constructive comments. 

I strongly agree with the questions you raised. Experimental task we used in the study could not be completely achieved the previous purpose that we want to investigate whether WN also induced disgusted emotion. In the new version, the purpose of the study was changed to exploring the different neural mechanisms between WD and WN by using affective priming effect. The abstract was adjusted. At the same time, two fMRI studies cited were modified to make them more consistent with the original paper. In the discussion section, we describe the results more objectively and are cautious to infer the implication of results.

---

## [Editor Report · Decision Letter 2]

1 Dec 2020

Different influences of moral violation with and without physical impurity on face processing: An Event-Related Potentials Study

PONE-D-20-17995R2

Dear Dr. Peng,

We’re pleased to inform you that your manuscript has been judged scientifically suitable for publication and will be formally accepted for publication once it meets all outstanding technical requirements.

Kind regards,

Natalie J. Shook

Academic Editor

PLOS ONE
---

## [Editor Report · Acceptance letter]

4 Dec 2020

PONE-D-20-17995R2 

Different influences of moral violation with and without physical impurity on face processing: An Event-Related Potentials Study 

Dear Dr. Peng:

I'm pleased to inform you that your manuscript has been deemed suitable for publication in PLOS ONE. Congratulations! Your manuscript is now with our production department. 

Kind regards, 

on behalf of

Dr. Natalie J. Shook 

Academic Editor

PLOS ONE